# Linking the Landscape of *MYH9*-Related Diseases to the Molecular Mechanisms that Control Non-Muscle Myosin II-A Function in Cells

**DOI:** 10.3390/cells9061458

**Published:** 2020-06-12

**Authors:** Gloria Asensio-Juárez, Clara Llorente-González, Miguel Vicente-Manzanares

**Affiliations:** Molecular Mechanisms Program, Centro de Investigación del Cáncer and Instituto de Biología Molecular y Celular del Cáncer, Consejo Superior de Investigaciones Científicas (CSIC)-University of Salamanca, 37007 Salamanca, Spain; gloriaasensiojuarez@gmail.com (G.A.-J.); clara.llorente@usal.es (C.L.-G.)

**Keywords:** *MYH9*, myosin, actin, macrothrombocytopenia, filament, motor activity, polymerization

## Abstract

The *MYH9* gene encodes the heavy chain (MHCII) of non-muscle myosin II A (NMII-A). This is an actin-binding molecular motor essential for development that participates in many crucial cellular processes such as adhesion, cell migration, cytokinesis and polarization, maintenance of cell shape and signal transduction. Several types of mutations in the *MYH9* gene cause an array of autosomal dominant disorders, globally known as *MYH9*-related diseases (*MYH9*-RD). These include May-Hegglin anomaly (MHA), Epstein syndrome (EPS), Fechtner syndrome (FTS) and Sebastian platelet syndrome (SPS). Although caused by different *MYH9* mutations, all patients present macrothrombocytopenia, but may later display other pathologies, including loss of hearing, renal failure and presenile cataracts. The correlation between the molecular and cellular effects of the different mutations and clinical presentation are beginning to be established. In this review, we correlate the defects that *MYH9* mutations cause at a molecular and cellular level (for example, deficient filament formation, altered ATPase activity or actin-binding) with the clinical presentation of the syndromes in human patients. We address why these syndromes are tissue restricted, and the existence of possible compensatory mechanisms, including residual activity of mutant NMII-A and/or the formation of heteropolymers or co-polymers with other NMII isoforms.

## 1. Introduction

Myosins constitute a large superfamily of proteins that convert the energy produced during ATP hydrolysis into a conformational change that propels molecular motion. In particular, class II myosins form filaments and create force and tension, because their motor domain binds to actin filaments. Class II myosins can be roughly divided into muscle (skeletal and cardiac), smooth muscle and non-muscle. There are three variants of non-muscle class II myosin, defined by their heavy chain. One of them, MHCII-A, is encoded in the *MYH9* gene. Several mutations of the *MYH9* gene in humans result in a wide spectrum of autosomal dominant disorders, jointly termed *MYH9*-related diseases (*MYH9*-RD). The main feature of *MYH9*-RD is congenital thrombocytopenia, along with platelet macrocytosis and the appearance of MHCII-A protein inclusions in the cytoplasm of neutrophils [1,2]. Four different syndromes are classified within this group of diseases: May-Hegglin anomaly (MHA), Epstein syndrome (EPS), Fechtner syndrome (FTS) and Sebastian platelet syndrome (SPS). They all feature early-age macrothrombocytopenia, characterized by low numbers of platelets of larger-than-normal size, which often increases bleeding. However, patients may develop additional clinical manifestations later in life, including presenile cataracts, loss of hearing and glomerular nephropathy. The condition of many of these patients worsen over time. For example, patients that only display macrothrombocytopenia early in their lives may develop nephropathy in their later years. However, the initial symptoms are often mild, thus many young patients remain undiagnosed. This is the reason why these disorders, which are catalogued as rare diseases, may be more prevalent than actually diagnosed. Current prevalence is established at approximately 3:1,000,000 [3,4,5]. Despite the wealth of clinical and genetic data, few attempts have been made to explain the phenotype of these patients based on the well-characterized cellular and molecular function of the *MYH9* gene product, MHCII-A, and its functional unit, NMII-A (see Section 2 for an explanation on nomenclature). This review intends to describe the current state of the art regarding the correlation between genotype and molecular phenotype from the point of view of a cell biologist, thus trying to fill this gap.

## 2. Organization of the *MYH9* Gene, MHCII-A Protein and NMII-A Hexamer

The *MYH9* gene is located on chromosome 22 q12-13 in humans. It contains 41 exons spanning almost 107 kbp (Figure 1). The first exon is not translated. Exons 2 to 41 encode a 1960 amino acid protein with an unmodified molecular weight of 226.59 kD. Throughout the review, we use *MYH9* to define the gene itself, MHCII-A for the protein product (encoded in the *MYH9* gene) and NMII-A for the fully formed, functional myosin II hexamer containing two MHCII-A chains, two essential light chains (encoded by the *MYL6* gene) and two regulatory light chains (encoded by the *MYL12A/B* genes). There are several variants of MHCII (myosin heavy chain) encoded in different genes. These are MHCII-B (encoded by gene *MYH10*) and MHCII-C (encoded by gene *MYH14*). The organization of the *MYH10* and *MYH14* genes is virtually identical to that of *MYH9*, indicating that the three genes evolved from ancestral duplication [6]. For nomenclature purposes, we refer to the protein products of the *MYH 9/10/14* genes (i.e., MHCII-A, II-B, II-C) as isoforms. We refer to the fully formed hexamers (NMII-A, NMII-B, NMII-C), including their light chains (ELC and RLC are common to all the isoforms) as paralogs. 

The motor domain of MHCII is located in the N-terminus. It contains the actin binding site and the ATP hydrolysis domains. It is encoded by exons 2–19. Exons 19 and 20 encode the binding region of myosin light chains. This structure, sometimes referred to as the “neck”, pivots to transform the force produced by the motor domain into movement. The coiled coil region is encoded by exons 21–40. This region mediates MHCII dimerization to form the NMII hexamer, which is the basic unit that forms myosin filaments (see Section 5 and Section 6). Finally, exon 41 encodes the non-helical “tailpiece”, C-terminal sequence (NHT). This region is highly divergent among isoforms and regulates filament formation through protein–protein interactions and/or phosphorylation [7]. This region has been used to generate isoform-specific antibodies that have revealed much information regarding their localization and function in live cells [6,8,9]. 

The basic structure of the NMII hexamer has been described in detail in many reviews (for example, [7,10,11]). The description here will be thus very brief. When extended (Section 5), the NMII hexamer looks like two little grapes attached to a single rod [12]. The N-terminus is globular (the two grape-shaped domains) as examined using rotary shadowing electron microscopy. The two light chains bind to its neck domain. The ELC binds to an IQ motif closer to the N-terminus; the RLC binds to an adjacent IQ-like motif next to it, but closer to the coiled coil. Two MHCII chains form a dimer through the coiled coil domain (the single rod domain). The rods can interact with the rods of other NMII hexamers in a parallel or anti-parallel fashion to form bipolar filaments. These act as nucleation sites for the formation of larger order mini-filaments, containing 20–30 hexamers each of approximately 300 nm [13,14].

NMII-A has relatively low kinetics of ATPase hydrolysis, particularly compared to muscle myosin II [15]. Its duty ratio (the time it remains bound to actin filaments) and the stability of NMII-A mini-filaments are also low (reviewed in [11]), indicating that NMII-A is well designed for quick processes that require small-scale forces and dynamic filament assembly and disassembly, for example in migrating cells. Visualization of GFP-MHCII-A in live cells revealed continuous assembly and disassembly during membrane protrusion [16]. The other two paralogs, NMII-B and NMII-C are kinetically better suited for force, strain and structure buildup in non-muscle cells [17,18]. As such, their assembly and turnover are much slower [19,20]. 

MHCII-A expression is ubiquitous, being detected in virtually every mammalian cell and tissue [21], including all embryonic stages [22]. In some cell types, it is the only isoform expressed, for example megakaryocytes and platelets, and neutrophils. Its levels are very low in mature neurons [23]. Both NMII-A and -B paralogs participate in the early stages of myofibrillogenesis [24,25], although their expression levels are also low in adult skeletal muscle [21] and absent in in mature cardiomyocytes [26] and the monkey cell line COS7 [27]. 

## 3. Clinical Features of *MYH9*-RD Patients

Most *MYH9*-RD patients feature congenital macrothrombocytopenia, and a few display non-syndromic deafness, nephritis and cataracts. There are several excellent reviews in the field (for example, [1,2]). Here, we have just summarized some of the clinical features to support later discussion of the possible correlation between phenotype and the molecular features of the *MYH9* mutants. 

### 3.1. Macrothrombocytopenia

This pathology is characterized by the presence of larger-than-usual platelets. Size does not imply per se that the platelets are dysfunctional. However, patients display low numbers of these large platelets. The precise reason for this phenotype is unknown (see Section 7). Interestingly, platelet count remains relatively stable throughout their lives [1]. In most cases, decreased numbers of platelets correlate with mild bleeding. Only 28% of patients suffer menorrhagia (severe menstrual bleeding), epistaxis (nasal bleeding), spontaneous mucosal bleeding and gum bleeding [3]. 

Platelets are generated in the bone marrow and are excised from giant, multinucleated precursors termed megakaryocytes (MKs). MKs differentiate from hematopoietic stem cells in the endosteal niche (closer to the inner surface of the bone). Once MKs mature, they migrate to the sinusoid region of the bone marrow (the center of the medulla) in response to CXCL12 (a chemokine formerly known as stromal cell-derived factor-1), and they extend protrusions into the sinusoids [28]. These protrusions are excised to form mature platelets by a combination of shear stress [29,30] and NMII-driven constriction [31] (Figure 2A). The only NMII paralog expressed in MKs is NMII-A. It normally localizes to the center of the cell and the peripheral granular zone. The fact that MHCII-A is the only isoform expressed in MKs is likely the cause of this feature of *MYH9*-RD. Reconstitution of some *MYH9*-RD mutations in mice has shown that MKs display chemotaxis defects that are related to their inability to properly activate NMII-A [32]. 

*MYH9*-RD patients are differentially diagnosed based on the size of their platelets. However, automated platelet counting does not recognize giant platelets, often leading to misdiagnosis of these syndromes. To distinguish *MYH9*-RD from other platelet syndromes, hematologists look for the presence of basophilic protein inclusions in neutrophils. These are termed Döhle-like bodies and mainly contain NMII-A (see Section 5.2). They are observed in a large percentage (42–84%) of *MYH9*-RD patients [5]. The range is so wide due to the varied levels of phenotypic severity. The presence of these inclusions help distinguish *MYH9* anomalies from other hematologic abnormalities with high specificity and sensitivity [35]. 

A syndrome with these features (leukocytic inclusions and the presence of giant platelets) was first described by May in 1909, and later by Hegglin in 1945. In 1972, Epstein et al. described a new syndrome characterized by hereditary nephritis, deafness and macrothrombocytopenia. This disease, severe in females, featured albuminuria (excess urine albumin) and microhematuria (erythrocytes in the urine) related to renal damage. Fetchner syndrome (FTS) was described in 1985 as a new variant of hereditary nephritis that also featured early onset cataracts and leukocytic inclusions. Sebastian platelet syndrome (SPS) was first described in 1990. SPS featured macrothrombocytopenia and different inclusions. The clinical and genetic manifestations of these specific syndromes have been the subject of several excellent studies and reviews [1,2,5,36].

### 3.2. Non-Syndromic Sensorineural Deafness

Not every *MYH9* mutation causes hematological manifestations. There is a specific type of non-syndromic sensorineural autosomal dominant hearing loss (DFNA17) also caused by *MYH9* mutations [37]. Loss of hearing is a common late-onset condition observed in *MYH9*-RD patients, with around 60% of subjects displaying impaired hearing late in their lives [38]. 

### 3.3. Nephritis 

Nephritis appears in approximately 25% of *MYH9*-RD patients. It is usually very aggressive, and it initially features proteinuria, sometimes with microhematuria, and rapid progression to chronic kidney disease that often requires transplantation [2]. The most severe cases of nephrotic failure have been described as coming from patients with motor domain mutations of MHCII-A [39,40].

### 3.4. Presenile Cataracts

Early-onset cataracts are the rarest feature of *MYH9*-RD patients [41]. As such, the correlation between the precise mutation and the appearance of this manifestation is currently unknown, likely due to its low prevalence. 

### 3.5. Other Manifestations of MYH-RD

Some *MYH9*-RD patients display alterations in liver enzymatic levels, mainly transaminases and γ-glutamyl transferase. This manifestation is quite benign, with few reported cases of liver failure [42]. 

## 4. Prevalent *MYH9* Genotypes

*MYH9*-RD include 49 different alterations affecting 12 exons of the *MYH9* gene [43]. Missense mutations are the most common, affecting 21 of the 1960 codons of the gene. A stop mutation has been detected in the last exon, generating a truncated MHC-IIA protein. Duplications and in-frame deletions are the least frequent, and are usually due to the introduction of repeated sequences [44]. Despite all this, between 20–35% of *MYH9*-RD cases present de novo mutations. Somatic or germinal mosaicism is rare [45].

In approximately 80% of the families, mutations are concentrated in codons for six specific amino acids: Ser96 (6%) and Arg702 (24%) located in the head domain, Arg1165 (9%), Asp1424 (20%), and Glu1841 (22%) in the coiled coil, and Arg1933 (19%) in the non-helical tailpiece (Figure 1) [44]. In this manner, most *MYH9*-RD cases are associated with a limited number of genotypes. These mutations have been identified in 77 unrelated families, with 22 different genotypes. Therefore, despite the fact that the *MYH9*-RD present diverse symptoms, they sometime share common mutations [36].

The main intent of this review is to correlate the phenotypes observed with the molecular defects caused by the mutations. To do this, it is important to acknowledge that the most crucial feature of the mutations is their location. Previous work has showed that patients with mutations in the exons encoding the motor domain had more severe thrombocytopenia and are at higher risk of developing nephropathy and deafness than patients with mutations affecting the tail domain or non-helical tailpiece [3,4]. However, there are exceptions, with patients bearing specific mutations in the tail domain also displaying extra-hematological features [46]. 

The most frequent mutations in the motor domain affect the Arg702 codon (Figure 1). These mutations (to Cys, His or Ser) cause the most complicated phenotype observed in *MYH9*-RD patients. Patients with R702 mutations are characterized by severe thrombocytopenia and are expected to display aggressive nephropathy and/or progressive deafness. 

Mutations in codons for Arg1165 and Asp1424 also increase the risk of developing extra-hematological complications. Patients bearing the D1424H missense mutation are expected to develop proteinuria and deafness by the sixth decade of life, and have a higher probability of developing cataracts. 

## 5. Regulation of NMII-A Activity

Myosin II is one of the most tightly regulated molecules in eukaryotic cells. This is likely due to its critical importance as the major generator of mechanical forces inside cells. This regulation takes place across multiple levels, including changes to the conformation; control of the ATPase activity and/or actin binding; formation of bipolar filaments; and growth, stability and disassembly of larger order mini-filaments.

### 5.1. Regulation of the Conforzmation of NMII-A and Its ATPase Activity

The best characterized regulatory mechanism of myosin II is common to smooth muscle (SMM) and non-muscle myosin II (NMII) paralogs. It consists of a conformational switch upon RLC phosphorylation. RLC phosphorylation in Ser19 extends the hexamer, enabling it to interact laterally with other extended NMII hexamers to form filaments [47]. It also increases the actin-dependent ATPase activity of the molecule [48]. In vitro studies using SMM, which is very similar to NMII, shows that non-phosphorylated SMM folds into a compact conformation termed 10S, which refers to its sedimentation coefficient [49]. 10S is commonly represented as a head-to-tail interaction that includes asymmetric head–head interactions that block the actin binding and ATPase domain [50,51]. In this conformation, the tail folds into three segments of similar length and winds around the heads [51]. Tail interactions with the head, with RLC and with itself further stabilize this inhibitory conformation [52,53]. Phosphorylation of RLC on S19 promotes the conformational extension (“opening”) of the molecule (conformation 6S), which becomes assembly competent and thus able to incorporate into filaments [12,54]. Phosphorylation of RLC on S19 seems to be a two-step process, in which both RLC chains have to be phosphorylated to break RLC–tail interactions [55] and extend the molecule [52,53]. Different kinases may phosphorylate RLC on Ser19 (reviewed in [56]), most notably myosin light chain kinase [57] and Rho-associated kinase (ROCK) [58]. Of note, ROCK seems to preferentially phosphorylate NMII-A-associated RLCs than those associated to NMII-B [59]. Although ROCK can phosphorylate RLC on Ser19 directly [58], it mainly acts through the phosphorylation and inactivation of MYPT1, the catalytic subunit of an RLC phosphatase [60]. Additional phosphorylation of Thr18 contributes to hexamer extension [61], increases myosin II actin-dependent ATPase activity [62] and favors the stabilization of NMII filaments, which controls cell polarity and adhesion dynamics [19,63]. 

The interconversion between these two conformations (10S ↔ 6S) regulates NMII filament distribution in vivo [64,65]. Folded 10S conformation likely enables the long-range diffusion of myosin, which can be incorporated into other subcellular compartments if/when needed [64]. In that sense, in vitro experiments have also shown that myosin monomers can form folded antiparallel oligomers of two or more molecules before extending into an assembly competent form [66]. These data suggest that, in a cellular context, this mechanism could not only apply to monomers, but also to myosin oligomers. These complexes could switch back and forth between a diffusible, assembly incompetent form and an assembly competent form that could be rapidly recycled and assembled upon RLC phosphorylation. 

Other post-translational modifications of RLC also regulate NMII ability to form filaments. Phosphorylation of Ser1 and Ser2 by PKC inhibits the function of NMII by decreasing the binding of NMII to actin filaments [67,68]. In platelets, this phosphorylation is necessary for NMII-A disassembly and reorganization in response to platelet-derived growth factor [68]. In mesenchymal cells, this mechanism prevents actomyosin assembly at the leading edge during chemotaxis [69]. Finally, phosphorylation of Tyr155 has been recently described to regulate NMII assembly through a different mechanism, which controls the formation of the NMII hexamer by impairing the interaction of the RLC with the heavy chain when Tyr155 is phosphorylated [70]. 

### 5.2. Regulation of Bipolar Filament Formation and Higher Order Mini-Filament Assembly and Disassembly 

NMII dimerization and bundling into higher order mini filaments occurs through electrostatic interactions between alternatively charged regions along the tail of the hexamer. Specifically, there is a critical and highly conserved positively charged region, the “assembly critical domain” (ACD) at the end of the coiled coil domain (Figure 1). The ACD can interact with negatively charged regions distributed along the tail in other NMII molecules [71,72]. Experimental data as well as modeling support the notion that NMII establishes electrostatic interactions to form parallel and anti-parallel arrays with well-defined periodicity. This is due to the amino acid sequence of specific regions within the coiled coil domain of NMII [73,74,75]. The current thinking is that, upon RLC phosphorylation, the NMII hexamer adopts a 6S conformation, which is highly unstable. Thus, it either increases its stability by forming these electrostatic bonds with similarly open 6S forms, or it rapidly folds back into the more stable 10S form. 

Classical immunofluorescence examination of NMII isoforms across cells and tissues has contributed to the notion that filaments are relatively stable. However, quantitative microscopy approaches such as fluorescence recovery after photo bleaching have shown that filaments rapidly exchange NMII hexamers, that is, hexamers from the filaments are replaced by hexamers that are in non-filamentous form in the cytoplasm [76], which must rapidly undergo the 10 → 6S conversion. Filaments also undergo RLC exchange. Exchanged RLC may be part of NMII hexamer turnover, or it may be released from the NMII hexamer and replaced with free, cytoplasmic RLC. Although direct evidence is lacking, RLC is likely to exist unassociated to NMII, either free or bound to proteins other than NMII. This is supported by the fact that RLC has additional binding partners, e.g., Myo18, calponin or NMDA receptors [77,78,79]. Other more indirect proof also supports the existence of free RLC. For instance, MHCII binds acidic liposomes through its RLC binding site, promoting the dissociation of the RLC [80], which likely becomes free. Thus, it is likely that photobleaching of RLC in filaments probably measures both components (NMII exchange in and out of filaments; and RLC in and out of NMII hexamers), particularly in light of the fact that the fractional recovery of fluorescent RLC (> 80%) is higher than those of any of the MHCII isoforms (≈ 55% for MHCII-A; ≈ 30% for MHCII-B and MHCII-C) [20,76].

The coiled coil domain that enables dimerization is followed by a non-helical domain that constitutes the tailpiece of the hexamer. Sequence alignment reveals that this is the most diverse region among NMII and SMM heavy chains. Homology in this region is almost non-existent, whereas it is up to 90% in the motor and coiled coil domains. Additionally, electrostatic studies indicate that the tailpiece does not dimerize, but likely remains untethered from the main NMII structure. Importantly, the end of the coiled coil and the non-helical tailpiece account for paralog-specific regulation. Phosphorylation and/or binding to regulatory proteins affect the establishment and/or maintenance of the aforementioned electrostatic interactions between the ACD domains. When these events occur in filamentous hexamers, they generate filament instability, promoting filament disassembly and thus contributing to cytoskeletal reorganization and plasticity.

Several kinases phosphorylate NMII-A (and II-B and II-C) on the last portion of the coiled coil domain and/or the non-helical tailpieces. These kinases include protein kinase C (PKC), transient receptor potential melastatin 7 (TRPM7) and casein kinase II (CKII) [72,81,82]. Due to the large divergence between isoforms in these regions, different residues have specific functions. Here, we only discuss NMII-A, not NMII-B or II-C. Ser1917 close to the ACD of NMII-A (Figure 1). Its phosphorylation by PKC controls the role of NMII-A in mastocyte degranulation [83]. Ser1943, located in the NHT domain, is phosphorylated by CK-II. Ser1943 phosphorylation dictates the level of NMII-A incorporated into filaments and the organization of NMII-A assemblies [84,85]. In mesenchymal stem cells, Ser1943 acts as a mechanosensor, becoming non-phosphorylated when cells are plated on stiffer substrates. This increases NMII-A assembly and enables NMII-B-dependent migratory polarization [86]. Importantly, Ser1943 phosphorylation determines NMII-A interaction with S100A4/Mts1 [85], which is a low molecular weight calcium sensor involved in metastasis [87]. Although the S100A4 binding site on MHCII-A does not include Ser1943, its phosphorylation is sufficient to prevent S100A4 interaction with NMII-A [85]. Calcium binding produces a conformational change that expose the NMII-A binding site, and the interaction with the end of NMII-A coiled coil prevents NMII-A assembly into filaments. Cells deprived of S100A4 display an excess of NMII-A filaments that translate into deficient chemotaxis [88] and decreased podosome formation [89]. Conversely, overexpression of S100A4 promotes metastasis [90], likely due to increased cell migration and cytoskeletal plasticity. Several other proteins have been reported to bind and regulate NMII-A filament assembly, including lethal giant larvae, myosin-binding protein H and LIMCH1, and Myo18. For further review on these interactions, please refer to [3,72]. 

## 6. Three Hypotheses to Explain the Correlation between Genotype and Molecular Phenotype

Deletion of the *MYH9* gene has catastrophic consequences during development. Somatic deletion of the *MYH9* gene in mice impairs embryo development due to acute defects in the formation of the visceral endoderm [91]. This is due to impaired cell–cell adhesion, which is also observed in epithelial cell cultures in which MHCII-A is deleted [92]. None of mutations described in *MYH9*-RD patients cause a complete lack of NMII-A [3]. Instead, they generate a spectrum of defective NMII-A variants, including: (i) motor deficient variants, e.g., R702C/H/S; (ii) mutants with altered dimerization, which include variants displaying mutations that affect the coiled coil region (aa 841–1890); (iii) variants with altered oligomerization, that is, those bearing mutations affecting exons 40–41, which encode the end of the coiled coil domain and the non-helical tailpiece. 

In general, mutations affecting the motor domain cause more severe phenotypes than those located to the coiled coil or non-helical tail domains. Pecci and co-workers have performed a formidable job compiling the type and severity of the phenotypes that appear in patients displaying specific mutations [2,3]. Their work clearly reveals that *MYH9*-RD are highly heterogeneous in terms of clinical manifestations and severity. Why is that? In the absence of systematic functional data in various cellular and molecular contexts, three hypotheses emerge, which are represented in Figure 3: i) different mutations cause diverse molecular effects. Some mutants are functional enough to enable a “close to normal” NMII-A function in cellular terms; this includes the possibility of finding mutant/wild-type assemblies that work similarly to wild-type filaments as a whole; ii) different mutations cause various degrees of protein aggregation over time, with deleterious consequences that would largely depend on the dynamics of aggregation; iii) NMII-A, either wild-type or mutant, can copolymerize with other myosins, if expressed in the cell/tissue under analysis. It is almost certain that a combination of these hypotheses explains the heterogeneous clinical presentation and evolution of *MYH9*-RD patients, and the following paragraphs are devoted to describe some specifics regarding each hypothesis. 

### 6.1. A Continuum of MYH9 Mutations that Cause Graded Molecular Defects

Regarding motor mutations, different mutations induce functional perturbations that impair myosin II function to diverse degrees. This is best illustrated by considering that this domain controls NMII interaction with actin and force generation as explained in terms of the swinging cross-bridge model (reviewed in [93]). For example, the R702C mutation renders a NMII-A which displays approximately 25% of ATPase activity and in vitro actin filament motility [94]. The degree of residual ATPase activity of other mutants of the motor domain, or the effect of the mutation on one, or several steps of the swinging cross-bridge cycle (actin binding, nucleotide exchange, phosphate release, etc.) has yet to be characterized (Figure 3A). 

It is not precisely known why coiled coil mutations display less severe phenotypes. NMII-A dimers, similar to *Drosophila melanogaster* NMII [73], form parallel and anti-parallel arrays at different staggers [74]. This could potentially allow dimers to “skip” mutant regions by establishing electrostatic interactions that are less favorable in the case of the wild-type form, but more favorable in the case of the mutant (Figure 3B). For example, the D1424N is not in the motor domain, yet this mutation increased focal adhesion formation [32], which is a hallmark of increased contractility [76,95,96]. This mutant may display stronger lateral interactions than wild-type NMII-A, forming more stable filaments and thereby increasing contractility-dependent effects, for example focal adhesion maturation. 

Finally, the effect of mutations in the non-helical tail are harder to predict (Figure 3C). One residue in the non-helical tail domain of MHCII-A (S1943) impairs filament formation when phosphorylated [84]. This region also binds to S100A4 (Mts1), and its interaction negatively regulates filament formation as well [85,97]. Egelhoff and co-workers generated a mutant form of MHCII-A in which the non-helical domain was deleted, and showed that such mutant myosin appeared over assembled [64]. However, whether mutations from *MYH9*-RD patients behave similarly is currently unknown. 

It is important to note that *MYH9*-RDs are genetically dominant diseases. This means that only one allele is mutated, and hence it is very likely that: i) mutant/wild-type heterodimers retain some degree of functionality, depending on the mutation; ii) there will be a significant proportion (at least 25%) of wild-type/wild-type homodimers, although the proportion of mini-filaments containing only wild-type NMII-A is very small; the bigger the filament, the smaller the probability. Cells from these patients would contain lower, but significant, levels of functional NMII-A. As an example, let us consider the case of a motor mutant that displays only 25% of actin/ATPase activity, but retains its ability to form filaments (this is the case of the R702C mutant, as described before [32,94]). Two assumptions are to be made for the sake of discussion: one is that there is no transcriptional regulation of the mutant allele because of the mutation. The other is that aggregation of the mutant protein is not taken into account (this is discussed in Section 6.2). In this case, the cell would contain 25% of NMII-A wild-type homodimers (100% functional); 50% of mutant/wild-type heterodimers (62.5% functional) and 25% of mutant homodimers (25% functional). Thus, and taking into account the assumptions made before, a cell bearing this mutation would contain 62.5% of the normal NMII-A activity found in wild-type cells (Figure 4A). This overall amount of functional NMII-A is likely sufficient to support most myosin-dependent cellular functions, except maybe those that exquisitely depend on fully activated NMII-A. This is likely the case of megakaryocyte excision into platelets at sinusoidal junctions (Figure 2B). In this regard, we have shown that the cellular phenotype caused by the shRNA-based depletion of NMII-A in fibroblasts is dose dependent, that is, the severity of the cellular phenotype is proportional to the degree of deletion, and cells displaying moderate decreases of NMII-A have relatively normal phenotypes (see Supplementary Figure 3 of ref. [16]). Conversely, the phenotype caused by NMII-B depletion is not as dose dependent, and a reduction of its levels below a certain threshold causes complete loss of cell polarity [19] and multinucleation [98]. This is likely the reason why mutations of *MYH10*, the gene encoding MHCII-B, are very rare [99]. Based on the onset of most *MYH9*-RD, it can be inferred that only a handful of cellular functions are dependent on the full activation of NMII-A. 

### 6.2. A Continuum of MYH9 Mutations that Causes Graded Protein Accumulation in Aggregates

A notable feature of *MYH9*-RD is that the majority of patients display early macrothrombocytopenia, but some of them display additional complications (kidney failure, deafness, presenile cataract, and elevation of liver enzymes) later in their lives [2]. This can be due to a number of reasons. One possibility is that some mutants tend to form basophilic aggregates, which are most apparent in granulocytes, but likely appear in other cell types. In fact, these aggregates are used to distinguish *MYH9*-RD from other platelet-related diseases. Staining with antibodies against MHCII-A revealed that most of these inclusions are made of NMII-A. The reason for these accumulations to appear are yet unclear, although it is unlikely that they are proteins misfolded during their biosynthesis, as these accumulations are not positive for aggresome markers (M. Vicente-Manzanares, unpublished observations). Aggresome markers are dyes that bind misfolded proteins when the ubiquitin–proteasome system is overwhelmed. In addition, most mutants appear to form part of NMII filaments at some point during their protein lifetime. These observations are hard to reconcile. One possibility is that genetic variants bearing *MYH9*-RD-causing mutations impair the 10S ↔ 6S conversion (Figure 4A,B). Due to the role of the head domain during the folding of the protein into the 10S, non-assembly competent form [100], some of these mutations could indeed affect the ability of the molecule to cycle between 10S and 6S states, causing the mutant to form aggregates in aberrantly folded forms (Figure 4A,B). These aggregates would nucleate the formation of the inclusion bodies observed in neutrophils [3,32,36]. This would also likely require that some of these mutations change the dynamics, or the molecular effects, of RLC phosphorylation in Ser19, which dictates this conformational change [101]. However, there is also evidence that motor mutants, or even IQ-deleted variants, can form filaments [32,64]. In this regard, an interesting hypothesis is that some of these mutations decrease the average number of times NMII-A is capable of folding back to the 10S state (and extend to the 6S state) before becoming irreversibly aggregated. This hypothesis would also be compatible with different mutations of the motor domain having different effects. Some mutations may increase the “fragility” of the whole molecule as it cycles between 10S and 6S, rendering it more prone to aggregate after a small number of conversion cycles, whereas other mutants would be more “robust”, bearing a larger number of 10S ↔ 6S cycles without undergoing aggregation. Aggregation could also be favored by additional mechanisms related to aging, such as post-translational modifications. Such mechanism would explain that, as individuals bearing *MYH9* mutations age, they become more prone to exhibit NMII-A aggregates, such as Döhle-like bodies. In addition, these aggregates may be responsible for renal damage. Cataracts would likely involve a dual mechanism that includes the accumulation of non-functional NMII-A protein in the eye leading to the crystalline becoming opaque, and deficient self-renewal of the epithelial cells that form the crystalline. This is part of the natural process of eye ageing and cataract onset that may be accelerated by *MYH9* mutations and subsequent protein aggregation. However, this remains to be determined experimentally. 

### 6.3. Heterodimerization and Copolymerization with Different Myosin Isoforms

Emerging evidence points to the ability of NMII-A to form heterodimers and co-polymerize with other NMII isoforms, particularly NMII-B and Myo18 [102,103,104]. A recent study suggests that A/B heterodimerization is favorable thermodynamically, opening the door for these heterodimers to have biological importance [74]. This is a clearly understudied part of the field regarding *MYH9* mutations, as some mutations, particularly those affecting the coiled coil and NHT domains, may have a profound impact on the dynamics of heterodimerization. NMII-B has a higher duty ratio, that is the time myosin expends tightly bound to actin (power stroke phase in Figure 3A), and a slower rate of contraction than NMII-A [105,106], which is consistent with slower filament turnover. If a given *MYH9* mutant product has a higher affinity for NMII-B than wild-type NMII-B, this would likely increase its permanence in filaments, since NMII-B has a much slower turnover in filaments than NMII-A [76]. A similar argument can be made for the formation of hybrid mini filaments made of mutant homodimers and NMII-B homodimers, or made of a mixture of homodimers and heterodimers. In all these contexts, NMII-B would be endowing the mutant NMII-A with additional filament stability and thus decreasing its probability to disengage from the filament and go back to the 10S state (Figure 4C). Each individual mutant is bound to have specific properties, further complicating the analysis. 

On the other hand, the role of Myo18 in this context remains unexplored. Myo18 co-assembles with NMII-A forming mixed filaments [103], promoting NMII stabilization into bundles [107]. However, Myo18 lacks enzymatic activity [79,108], hence its function is likely to be limited to nucleation and/or stabilization of preformed filaments. This was proposed early [103] and has been further suggested by electrostatic analysis of its interaction with NMII-A. As for its role in the context of *MYH9*-RD, it is currently unknown, but it is tempting to speculate that it could nucleate mutant NMII-A to favor its filamentation. It will be interesting to study whether mutant NMII-A aggregates include Myo18. 

The wide spectrum of heterodimerization and co-polymerization possibilities could endow compensatory effects in most cellular contexts, and justify that the clinical manifestations of these diseases are restricted to cells in which NMII-A is the only, or at least the prevalent, paralog present. However, there could also be molecular contexts in which the redundancy between paralogs is not enough and the unique properties of NMII-A are crucial. These properties are determined mostly by the dynamic properties of the head (the higher ATPase activity, lower duty ratio than the other paralogs) [109] but also by the tail, that determine cellular localization and filament stability [19,110,111]. 

## 7. Why do MYH9-RD Patients Display Few, but Giant, Platelets?

Despite extensive characterization of the genotype and clinical manifestations of *MYH9*-RD patients on one side, and the in-depth description of the molecular dynamics of NMII-A function in diverse contexts, both aspects are relatively disconnected. The intrinsic complexity of the system, in which the nature of the mutations—the amount of protein produced, its tendency to aggregate, and the role of other isoforms—makes it almost impossible to come with a satisfactory explanation for every clinical manifestation of the disorders. However, most *MYH9*-RD patients display macrothrombocytopenia. Can the molecular hypotheses formulated in the previous section explain this clinical manifestation?

As stated in Section 3, *MYH9*-RD MKs chemotax deficiently towards sinusoids. However, sinusoid juxtaposition is reduced, but not completely impaired [32]. Once adjacent to sinusoids, MKs form platelets by excision of protrusions inserted within the junctions of endothelial cells that line the sinusoids. Excision is favored by shear stress, but it also requires NMII-A constriction. The mechanism can be envisioned as similar to the formation of clathrin-coated pits, except in the other direction of the plasma membrane. Of note, clathrin-coated pit formation also requires NMII [112]. In a wild-type MK, NMII-A would provide constriction at the junction between the body of the MK and the proplatelet, and shear stress would provide the flow to elicit excision. A mutant NMII will likely localize properly (although in cases of extreme aggregation, even this step may be compromised). However, its attempt to provide sufficient constriction may fail, either because it does not retain its whole motor activity (motor mutants), or because it displays filament instability (rod mutants). In both cases, the outcome is an alteration of myosin processivity, which can be defined as the capacity of NMII to constrict actomyosin filaments for several cycles without releasing them. Though not needed for every NMII-related cellular function, “perfect” processivity may be fundamental in cellular processes such as excision, which requires that NMII orchestrates the application of high levels of contraction in a localized manner. A similar case is that of the formation of the cytokinesis ring. A recent study from the Burnette group has shown that NMII-A/B chimeras are directed to the cytokinesis ring through their tail domains, but the head domain swap decreases separation efficiency [111], lending support to the argument that full activation of one specific paralog may be required for some functions. In other instances, other fully competent myosin paralogs may compensate for this, if present. However, in the case of MKs, this surely contributes to excision failure since NMII-A is the only paralog present. 

The larger size of the platelets present in these patients can be explained as follows: let us consider a wild-type MK, which reaches the central zone of the bone marrow and starts injecting proplatelet material into a sinusoid. When the proplatelet reaches a threshold size, NMII-A-driven constriction plus the shear induced by the blood flow excises the proplatelet. In MKs bearing *MYH9* mutations, we propose that this process initially takes place in a similar manner (although less frequently, since fewer MKs reach the sinusoids due to deficient chemotaxis, as explained before). However, the net excising force generated by NMII-A constriction at the neck of the sinusoid plus shear stress does not reach the force threshold required to excise the proplatelet. The hypothesis is that the MK keeps injecting material independently of excision, thus swelling the proplatelet. A number of cycles of failed constriction + shear-driven excision would result in a giant proplatelet that bears great mechanical resistance to shear stress, which then may excise it even without the contribution of NMII-A-driven constriction. This mechanism, based on the molecular models of Section 6, could explain the enlarged size of the MYH9-RD platelets and their fewer numbers (Figure 2B). 

## 8. Concluding Remarks

*MYH9*-RD are rarely life-threatening unless they cause renal failure. In addition, patients do not appear to be predisposed to suffer from other, more severe diseases, such as cancer or neurodegenerative diseases. However, they present fascinating phenotypes with very specific disorders. The rather large number of mutations of the *MYH9* gene and their correlation with the sophisticated regulation of the function of NMII-A provides a wide spectrum of possible molecular defects, and current research has just scratched the surface. We expect that a concerted effort by structural biologists and biochemists will explain the precise molecular effect of each mutation on the structure of NMII-A in filamentous and non-filamentous states. These experiments, together with cell biology approaches to show how these structural defects translate into the actual function of the protein in cells and tissues, will shed light on the regulation of this protein. This is of crucial importance because NMII-A is a cornerstone of cell mechanics and a major generator of intracellular force and mechanosensing.

## Figures and Tables

**Figure 1 cells-09-01458-f001:**
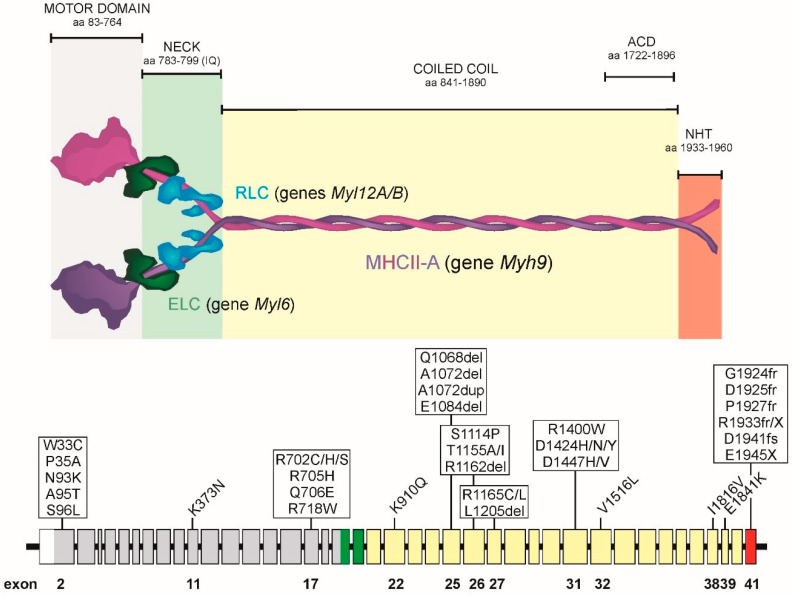
Top, Color-coded organization of an extended NMII-A hexamer, including the domain organization of MHCII-A (purple and magenta, different colors are used only to highlight the fact that they are two independent, yet equal, polypeptides), and their binding to ELC (in dark green) and RLC (blue). ACD, assembly competent domain. NHT, non-helical tailpiece. Bottom, exon organization of the *MYH9* gene and position of the most prevalent mutations found in MYH9-RD patients. Exon organization is also color coded: grey, motor domain; green, neck; yellow, coiled coil; red, NHT.

**Figure 2 cells-09-01458-f002:**
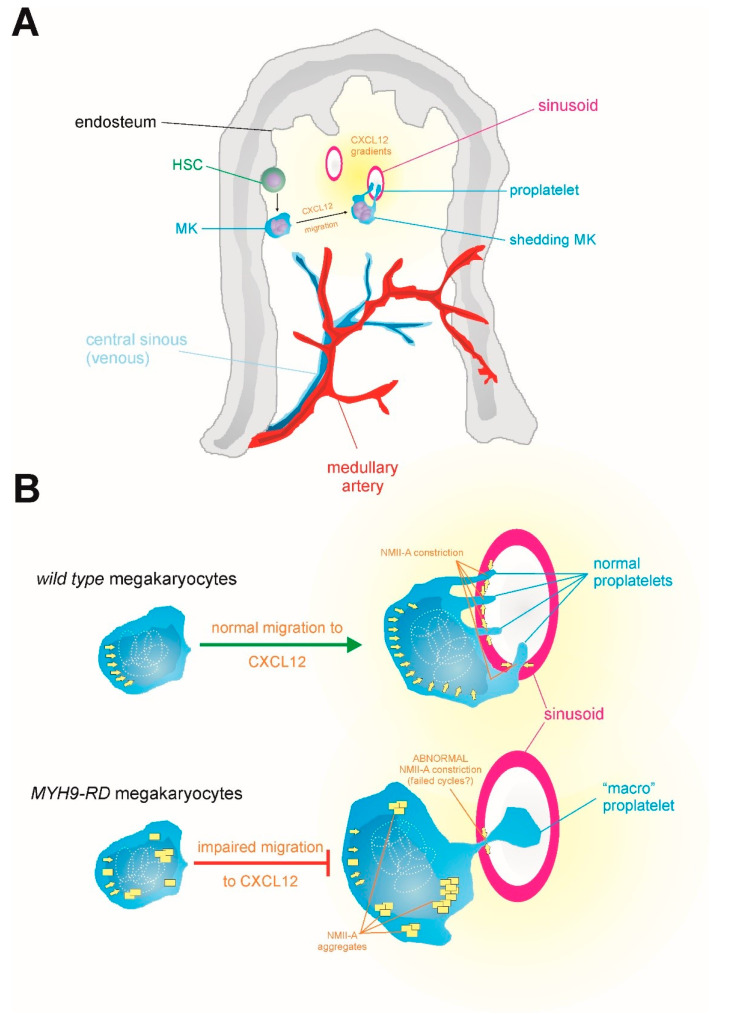
(**A**) Scheme depicting the generation and migration of megakaryocytes (MK) in the bone marrow. MK differentiate from HSC in the endosteum and then migrate towards the sinusoids driven by CXCL12 gradients. There, they insert protrusions in the gaps between endothelial cells lining the sinusoids. The protrusions become proplatelets. (**B**) Comparison between wild-type and MYH9-KD MKs. Ploidy (represented as white dotted ellipses) is normal in both cases, but MYH9-RD MKs migrate deficiently towards sinusoids, which depends on CXCL12 [32]. The combination of the shear stress driven by blood flow in the sinusoid and NMII-A-driven constriction excises the proplatelets (as explained in Section 3). The molecular reason for the onset of macrothrombocytopenia in MYH9RD patients has not been formally explained (see Section 7). The figure also integrates the known role of NMII-A to push the nucleus in three dimensions, represented by yellow arrows behind the multinucleated core. This has not been shown explicitly for MKs, but it has for other cells of hematopoietic origin, such as dendritic cells [33,34]. Yellow squares represent NMII-A aggregates in MYH9-KD MKs.

**Figure 3 cells-09-01458-f003:**
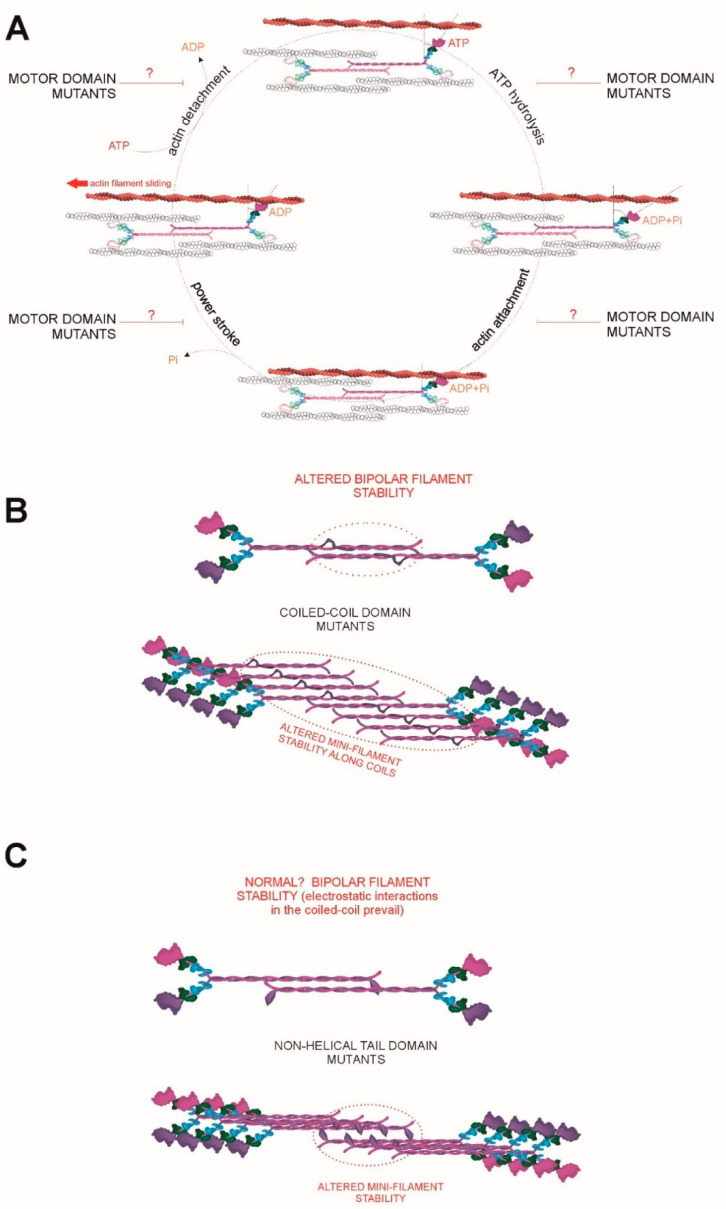
(**A**) Schematic representation of the cross-swinging cycle [93], and the possible effects of diverse motor mutants found in MYH9-RD patients. The mutations can affect ATP hydrolysis (top right), actin binding (bottom right), phosphate release (bottom left) and/or nucleotide exchange and/or actin detachment (top left). (**B**) Possible effects of mutations in the coiled coil domain (represented as a bulge in the center of the coil). Depending on the precise site and type of mutation, the formation of the initial bipolar mini-filament (top) may be impaired, or enhanced. The same argument applies to their growth into mini-filaments (bottom). (**C**) Possible effects of mutations in the NHT domain (represented as a bulked-up tail). It is unlikely that bipolar filament formation (top) is largely affected as electrostatic interactions between coiled coils dominate this step [73,74]. Color codes of the different chains is as in Figure 1.

**Figure 4 cells-09-01458-f004:**
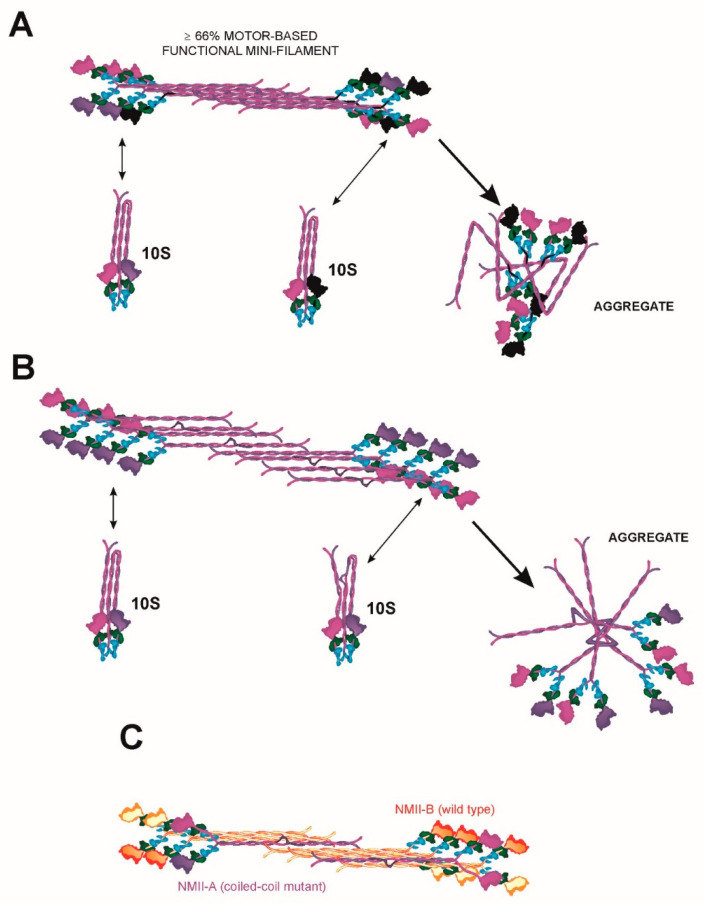
(**A**) Schematic representation of a mixed mini-filament including wild-type (magenta/purple) and a generic motor mutant (black) NMII-A. Functionality (≥ 66%) is calculated for the specific composition of the mini-filament (eight wild-type heads and four mutants). In this particular filament, functionality would be 66.6% if the mutant had 0% activity, as discussed in Section 6.1. Bottom left represents the exchange of wild-type NMII-A. Bottom right represents the folded conformation into three segments and the possibility that the mutant folds back into 10S normally, or does not fold back into 10S, aggregating in an aberrant conformation instead. (**B**) Schematic representation of a mixed mini-filament including wild-type (magenta/purple) and a generic coiled coil mutant (bulge in the coiled coil) NMII-A. Bottom left represents the exchange of wild-type NMII-A. Bottom right represents the possibility that the coiled coil mutant folds back into 10S, or cannot fold back into 10S, aggregating in an aberrant conformation instead. (**C**) Cartoon depicts the integration of NMII-A, wild-type (magenta/purple) or mutant (black) into mini-filaments with other NMII paralogs, e.g., NMII-B (in orange/red tones), which is thermodynamically feasible, as shown in [74]. Color codes of the different chains is as in Figure 1.

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
