# Peer review of "Linking the Landscape of MYH9-Related Diseases to the Molecular Mechanisms that Control Non-Muscle Myosin II-A Function in Cells"

_cells, 2020, doi:10.3390/cells9061458_

Round 1
Reviewer 1 Report
This is an excellent review on mutations in the gene encoding non-muscle myosin IIA and on the relation of these mutations to human diseases. I enjoyed the informative text, the critical discussion, and the clear writing.
There are a couple of minor points I like to address.
Lines 106-110: What about expression of MHCII-A in skeletal muscle?
Line 131: What is CXCL12? A list of abbreviations would be helpful.
Line 144: What is the reason for the large variation of percentages?
Line 157: “However” does not seem to fit.
Line 171: “correlation phenotype – genotype”: the meaning should be specified.
Line 181: “alterations affecting 14 exons”. In Fig.1, I count 12 exons affected. The remaining two might also be shown.
Line 191: 77 families – 22 different cases of MYH9-RD. This is not clear to me: does it mean 22 different genotypes?
Line 224: Again, explain CXCL12.
Lines 226-227: In the figure, I can’t see the nuclei pushed. What is the faint, dotted network?
Lines 235-236 (and 250): Is this actin-dependent ATPase activity?
Lines 238-243: It may be helpful to show part of Fig. 4A here and to focus later in Fig. 4 on aggregate formation.
Line 255-256: “…can form folded…oligomers…before unfolding”. Is “unfolding” the right term or “extending” into the 6S state?
Line 263: Delete “in”.
Line 276: It would be helpful to indicate the ACD in Fig. 1 and to refer here and on line 316 to this figure.
Line 323: explain S100A4.
Line 358: …explains…
Line 375: Kauffman is Kaufmann in reference 71.
Lines 376-378: I don’t understand the argument on head splaying and separation of the mutated region.
Line 409: …caused by deletion… (Incidentally, is the gene or the protein deleted?)
Line 440: …is yet…
Line 441: What are the aggresome markers, which would make mis-folding likely?
Line 488: What does “homeostatic” mean in this context of a phosphorylation-dependent change?
Line 452: What are IQ-deficient variants?
Line 460: Delete “etc.” or specify.
Line 465: “deficient cell migration”: which migrating cells are relevant?
Line 477: Please explain the argument that a higher duty ratio suggests slower filament turnover.
Line 478: What is the “also” related to?
Lines 505-506: The preference for large staggers and central positions of Myo18 is not obvious to me. If I read reference [71] correctly, the preference holds only for anti-parallel staggers.
Line 511: Please extend on the molecular context that explains redundancy.
Line 528: Replace “still” probably by “also”.
Line 535: “displays”.
Line 539: There appears to me a relationship to myoII function in cytokinetic ring formation. Is this relevant?
Lines 546-547: I don’t understand the sentence. Does it implicate that pro-platelets fuse when compressed?
Line 555: “…are rarely severe”. However, on line 202 it is stated that Arg702 mutations, the most frequent ones, cause the most severe phenotype.
Fig. 4: Explain blue color (light chains).
Author Response
Reviewer 1-
This is an excellent review on mutations in the gene encoding non-muscle myosin IIA and on the relation of these mutations to human diseases. I enjoyed the informative text, the critical discussion, and the clear writing.
Many thanks for your kind appraisal of the work.
There are a couple of minor points I like to address.
- Lines 106-110: What about expression of MHCII-A in skeletal muscle?
Thanks for the question. Reference 21 (Golomb et al., 2004) reports low, but significant, expression of NMII-A in skeletal muscle. Also the work from Sanger describes how NMII-A nucleates muscle myosin II to form sarcomeres (references 24 and 25). This is now in lines 119-121.
- Line 131: What is CXCL12? A list of abbreviations would be helpful.
CXCL12 is defined as stromal cell-derived factor with chemotactic properties (chemokine). We have included a list of abbreviations.
- Line 144: What is the reason for the large variation of percentages?
This is due to the wildly different phenotypic differences between patients. We have incorporated this in the discussion of this sentence (line 157).
- Line 157: “However” does not seem to fit.
We have rephrased this sentence. Thank you.
- Line 171: “correlation phenotype – genotype: the meaning should be specified.
We have rephrased this sentence to clarify it. Thank you (lines 189-191).
- Line 181: “alterations affecting 14 exons”. In Fig.1, I count 12 exons affected. The remaining two might also be shown (line 197).
Thanks, this was a typo. It is 12 exons. It is now corrected (line 201).
- Line 191: 77 families – 22 different cases of MYH9-RD. This is not clear to me: does it mean 22 different genotypes?
The reviewer is correct. 22 different genotypes. Changed to “22 different genotypes” as indicated in line 207.
- Line 224: Again, explain CXCL12.
This is a chemokine and it is now included in the list of abbreviations. Unfortunately, the Zlotnik nomenclature CXCLxx is not very illuminating as to the function of the different chemokines.
- Lines 226-227: In the figure, I can’t see the nuclei pushed. What is the faint, dotted network?
This is represented by the yellow arrows behind the faint ellipses that represent the multinucleated core of MKs. Thanks for requesting this clarification, which we have incorporated into the figure legend (lines 235-238).
- Lines 235-236 (and 250): Is this actin-dependent ATPase activity?
Thanks for picking this up. It is actin-dependent and we have modified the sentence accordingly (line 255-256 and 272-273).
- Lines 238-243: It may be helpful to show part of Fig. 4A here and to focus later in Fig. 4 on aggregate formation.
This is a good suggestion that we have implemented by calling Figure 4A here, as requested (line 261).
- Line 255-256: “…can form folded…oligomers…before unfolding”. Is “unfolding” the right term or “extending” into the 6S state?
We have changed to “extending” as suggested (line 279).
- Line 263: Delete “in”.
Done as indicated.
- Line 276: It would be helpful to indicate the ACD in Fig. 1 and to refer here and on line 316 to this figure.
We have changed the figure as indicated, and call it in line 299 and 339-340.
- Line 323: explain S100A4.
We have added a brief explanation on the role of S100A4 and add a new reference (lines 348-349).
- Line 358: …explains…
Corrected in line 382.
- Line 375: Kauffman is Kaufmann in reference 71.
This sentence has actually been rephrased so the authors are not named anymore. Still, thanks for picking up the typo.
- Lines 376-378: I don’t understand the argument on head splaying and separation of the mutated region.
Due to the speculative nature of this sentence, we have opted to remove it.
- Line 409: …caused by deletion… (Incidentally, is the gene or the protein deleted?)
We have clarified this. These were done using shRNA, so it is mRNA and protein (lines 430-431). There was a “by” missing that has misled the reviewer. We apologize for it.
- Line 440: …is yet…
Corrected.
- Line 441: What are the aggresome markers, which would make mis-folding likely?
This has been clarified (lines 449-450). We think it is important that reviewer knows that the Aggresome identification kits are commercial and their composition is not available as they are patent-protected. It is speculated that they are dyes with a specific composition that enables them to bind to misfolded proteins.
- Line 488: What does “homeostatic” mean in this context of a phosphorylation-dependent change?
FRAP experiments from the many groups, including Adelstein, Ikebe, Means and ourselves, have shown that filaments undergo NMII exchange in “homeostatic” conditions, that is, in the absence of stimulus. However, for clarity we have opted to remove this word from the legend of figure 3.
- Line 452: What are IQ-deficient variants?
We have replaced “deficient” with “deleted” and added an explanation on the meaning of such deletion (line 461).
- Line 460: Delete “etc.” or specify.
Removed as indicated (line 470).
- Line 465: “deficient cell migration”: which migrating cells are relevant?
Thank you, we have rephrased this. Probably there is no migration, but self-renewal of epithelial cells (lines 476-477).
- Line 477: Please explain the argument that a higher duty ratio suggests slower filament turnover.
We have included such an explanation (lines 497-500). As the reviewer is likely to know, the turnover rate of NMII in filaments is mainly dominated by the ACD. The main argument that actin binding contributes to filament turnover can be based on the FRAP data that we published in 2007 using Ma and Adelstein’s R709C mutant of myosin II-B. This mutant has a severely impaired in vitro actin motility and a 3-fold reduced ATPase turnover ratio (hence a higher duty ratio as the mutant is very slow exchanging ADP for ATP) and a measurable slower turnover compared to wild type II-B. However, for clarity, we have replaced “suggesting” with “consistent with”.
- Line 478: What is the “also” related to?
“Also” has been removed.
- Lines 505-506: The preference for large staggers and central positions of Myo18 is not obvious to me. If I read reference [71] correctly, the preference holds only for anti-parallel staggers.
We agree and at this point, this point is speculative even in the brilliant Kaufmann and Schwarz study. Hence, we have limited the discussion on Myo18 to its possible role as a nucleator or stabilizer, which is well-established by different groups (lines 510-511).
- Line 511: Please extend on the molecular context that explains redundancy.
We have done so in lines 516-524.
- Line 528: Replace “still” probably by “also”.
Done as requested (line 555).
- Line 535: “displays”.
Changed as requested (line 562).
- Line 539: There appears to me a relationship to myoII function in cytokinetic ring formation. Is this relevant?
Yes it is. We have included this here along with recent evidence from the Burnette group showing that even well-localized chimeras do not rescue completely the ability of NMII to support cytokinesis (lines 568-570).
- Lines 546-547: I don’t understand the sentence. Does it implicate that pro-platelets fuse when compressed?
This is hypothetical, but we have rephrased it because it is worth exploring. The idea is, a wild type MK injects material into a sinusoid. When the protrusion (the proplatelet) reaches a threshold size, NMII-A + blood flow will excise the proplatelet. In a mutant, we propose that the process is similar, but the addition of NMII-A constriction + blood flow cannot excise the platelet. However, the MK keeps injecting material, thus the proplatelet swells. Do this a number of times and you end up with a giant proplatelet that bears great resistance to the blood flow, hence the flow may excise it even without the cooperation of the NMII-A-driven constriction. We believe the reviewer’s comment has made us rephrase this in a clearer and more exciting manner (lines 575-586).
- Line 555: “…are rarely severe”. However, on line 202 it is stated that Arg702 mutations, the most frequent ones, cause the most severe phenotype.
Thanks for picking this up. We meant “life-threatening”, and have changed the sentence to better convey that meaning (line 590).
- 4: Explain blue color (light chains).
Thanks for picking this up too. We have added the color codes to the legends of Figures 1, 3 and 4.
Reviewer 2 Report
This is an excellent review, which I enjoyed reading. I have a few minor points that the authors could address.
macrothrombocytopenia is introduced early in the introduction but not defined until section 3.1. A brief definition when this is first introduced could help to orient the reader early on?
line 104: should there be a comma between force and strain?
Author Response
This is an excellent review, which I enjoyed reading. I have a few minor points that the authors could address.
Many thanks for your positive evaluation.
2.1. Macrothrombocytopenia is introduced early in the introduction but not defined until section 3.1. A brief definition when this is first introduced could help to orient the reader early on?
We have now defined this in line 52-53.
2.2. Line 104: should there be a comma between force and strain?
Comma has been added (line 114).
Reviewer 3 Report
This is an insightful and comprehensive review of the function of MYH9, with a thorough and careful explanation of how the cell biology of this myosin might help us understand the clinical manifestations of myosinopathies. I found it generally easy to read and highly informative, and have only a few minor content and editorial suggestions.
Consider the audience for this review. If the audience is cell biologists, it would be helpful to include definitions of the probably unfamiliar clinical terms like macrothrombocytopenia, menhorragia, epistaxis, albuminuria, and microhematuria. If the audience is instead clinicians, then perhaps the cell biology terms may need a bit of explanation. As a cell biologist, I would leave it to a clinician to point out which cell biology or motor protein terms might be unfamiliar (although ‘duty ratio’ might be an example).
Line 248. Mention ROCK also phosphorylates and inhibits the myosin phosphatase.
Line 395. While 25% of dimers may be wildtype, it is vanishingly unlikely that the whole filament would be composed of wildtype subunits. This might be worth mentioning.
Remove filler phrases:
Line 49. Delete ‘it is considered that’
Line 66. Delete ‘it is noticeable that’
Line 101. Delete ‘This has been experimentally proven in experiments in which’
In addition to being an unnecessary phrase, ‘proven’ is problematic
Line 382. Delete ‘it can be argued that’.
Other minor edits:
Line 118. ‘to later discuss’ to support later discussion of
Line 122. No comma after size
Line 399. ‘there are no transcriptional regulation’ ‘there is no transcriptional regulation’.
Line 409. ‘phenotype caused BY the deletion’
Line 427. Odd spacing of line
Line 540. ‘may compensate FOR this’
Line 560. ‘a concerted effort BY structural biologists’
Author Response
This is an insightful and comprehensive review of the function of MYH9, with a thorough and careful explanation of how the cell biology of this myosin might help us understand the clinical manifestations of myosinopathies. I found it generally easy to read and highly informative, and have only a few minor content and editorial suggestions.
Consider the audience for this review.
- If the audience is cell biologists, it would be helpful to include definitions of the probably unfamiliar clinical terms like macrothrombocytopenia, menhorragia, epistaxis, albuminuria, and microhematuria. If the audience is instead clinicians, then perhaps the cell biology terms may need a bit of explanation. As a cell biologist, I would leave it to a clinician to point out which cell biology or motor protein terms might be unfamiliar (although ‘duty ratio’ might be an example).
Thanks for this comment. Actually, the idea is to bring the two fields together. To do this, we have defined the diseases and clinical manifestations in a clearer way and explain the biochemical and molecular concepts as clearly as possible. We are happy that the reviewer has picked up the spirit of the review.
- Line 248. Mention ROCK also phosphorylates and inhibits the myosin phosphatase.
We have done so and added the original reference (Kimura et al., 1996) in lines 269-271.
- Line 395. While 25% of dimers may be wildtype, it is vanishingly unlikely that the whole filament would be composed of wildtype subunits. This might be worth mentioning.
Done as suggested (line 414-416). In fact, the reviewer is absolutely correct, the bigger the filament is, the smaller the possibility. We have explicitly mentioned it.
- Line 49. Delete ‘it is considered that’
- Line 66. Delete ‘it is noticeable that’
- Line 101. Delete ‘This has been experimentally proven in experiments in which’
- In addition to being an unnecessary phrase, ‘proven’ is problematic
- Line 382. Delete ‘it can be argued that’
- Line 118. ‘to later discuss’ à to support later discussion of
- Line 122. No comma after size
- Line 399. ‘there are no transcriptional regulation’ à ‘there is no transcriptional regulation’.
- Line 409. ‘phenotype caused BY the deletion’
- Line 427. Odd spacing of line
- Line 540. ‘may compensate FOR this’
- Line 560. ‘a concerted effort BY structural biologists’
Thanks for all these style modifications, which we have incorporated into the final version of the manuscript.